# Weight Gain in Adults with Avoidant/Restrictive Food Intake Disorder Compared to Restrictive Anorexia Nervosa—Pilot Findings from a Longitudinal Study

**DOI:** 10.3390/nu13030871

**Published:** 2021-03-07

**Authors:** Magnus Fjeldstad, Torben Kvist, Magnus Sjögren

**Affiliations:** 1Psychiatric Center Ballerup, Maglevaenget 32, 2750 Ballerup, Denmark; magnus.fjeldstad@gmail.com (M.F.); torben.madsen.kvist.02@regionh.dk (T.K.); 2Institute for Clinical Medicine, Copenhagen University, 1165 Copenhagen, Denmark

**Keywords:** avoidant restrictive food intake disorder, ARFID, adults, anorexia nervosa, restrictive eating, eating disorders, weight restoration, inpatient setting

## Abstract

Background: Avoidant/Restrictive Food Intake Disorder (ARFID) is characterized by persistent failure to meet nutritional needs, absence of body image distortion and often low body weight. Weight restorative treatment in ARFID-adults is provided for as in Anorexia Nervosa (AN), while the effect is unknown. The aim was to compare weight gain between ARFID and restrictive subtype of AN (AN-R), including exploring impact of medical factors and psychopathology. Methods: Individuals with ARFID (*n* = 7; all cases enrolled over 5 years) and AN-R (*n* = 80) were recruited from the Prospective Longitudinal All-comers inclusion study in Eating Disorders (PROLED) during 5 years. All underwent weight restorative inpatient treatment. Clinical characteristics at baseline and weekly weight gain were recorded and compared. Results: There were no significant differences at baseline weight, nor in weight gain between groups. Anxiety was statistically significantly higher in AN-R at baseline. Conclusions: Although there were differences in several clinical measures at baseline (Autism Quotient, symptom checklist, mood scores and Morgan Russel Outcome Scale), only anxiety was higher in AN-R. No differences in weight gain were observed, although mean values indicate a faster weight gain in the ARFID group. Standard weight restorative treatment in this study in adults with ARFID has similar weight gaining effect as in AN-R.

## 1. Introduction

Avoidant Restrictive Food Intake disorder (ARFID) is an eating or feeding disorder characterized by, e.g., apparent lack of interest in food or eating, concerns about the aversive effects of eating, or avoidance of food based on the sensory characteristics of the food, and either weight loss, significant nutritional deficiency, dependence on supplements, or marked interference with social functioning [1]. Furthermore, nutritional intake and its effects are not caused by food unavailability, nor a manifestation of another medical or mental condition, and not due to substances affecting the central nervous system including withdrawals. Contrary to Anorexia Nervosa (AN), eating behaviour is not motivated by a preoccupation of body weight or shape. The point prevalence of ARFID has been reported to be 0.8% for women and 0.9% for men [2]. Comorbidities are common and potential life-threatening medical complications may occur and prompt early identification [2].

ARFID is a relatively newly described disorder and due to the low body weight and the restrictive food intake, may be mistaken for AN which is, in contrast to ARFID, characterized by phobia of weight increase, low body weight, body image distortion and compensatory behaviours such as frequent exercise. Few studies have examined the effects of weight restorative treatment in ARFID. We performed a systematic literature search using PUBMED on weight gain or weight increase in ARFID identified 19 studies whereof only 8 compared weight gain in ARFID to AN, and none included adults only but children to young adults (mean sample size 23.5 cases of total 188 ARFID cases). Some studies found a slower weight gain in ARFID compared to AN [3,4], and one study a lesser weight gain in ARFID [5], while a few other studies did not find any significant differences in weight gain [6,7,8]. Background or influencing factors were not systematically investigated and the follow-up time varied between 1 week [3,6] to 3–4 weeks [3,4] and 10 weeks [5,9,10] or more [11]. The rationale for a slower and lesser weight gain in ARFID may relate to the differences in psychopathology. Individuals with ARFID frequently share traits with autism spectrum disorder (ASD) [12,13,14], where rigid principles may interfere with treatment approaches that are developed from experience in treating AN such as promoting intrinsic motivation [15]. This may be reflected at baseline assessments, for example in readiness to change. This may also explain why individuals with ARFID more often have a chronically low weight compared to AN [16].

In view of the few studies executed investigating weight gain in ARFID, and the heterogeneity in terms of study design and treatments provided, there is a need for evaluating the effect of standard weight restorative treatment, given as successively increasing calorie content in daily meals to ensure an approximate 1 kg/week weight gain, typically provided to patients with AN, on weight gain in ARFID. Consequently, the aim of the current study was to compare the effect of weight restorative treatment in ARFID compared to AN-R, including exploring baseline factors that may influence weight gain during inpatient treatment. AN-R was selected for comparison since this is the diagnosis most similar to and most difficult to distinguish from ARFID. Any disparate findings may influence the development of future treatment programs for ARFID. We used data from the Prospective Longitudinal All-comer inclusion study on Eating Disorders (PROLED) in which a broad range of baseline factors are investigated. Our hypothesis was that individuals with ARFID would have a slower weight gain than those with AN-R.

## 2. Materials and Methods

### 2.1. Participants

Between January 2016 and December 2020, 7 patients with ARFID were identified using ICD-11 diagnostic criteria [1], together with 80 AN-R, in the PROLED study. The ongoing PROLED study is a clinical naturalistic 10-year annual follow-up study on eating disorders and is expected to have an annual intake of at least 100 AN patients per year. It has been approved by the local ethics board (id: H-15012537; addendum 77106) and the data processing board. The following individuals are eligible for enrollment subject to signed written informed consent:-adult individuals (age range 18–65 years)-admitted to the ED unit in Psychiatric Center Ballerup, Denmark-a diagnosis of an eating disorder

Subjects undergoing forced care were excluded from the study. The enrollment rate was 96% in 2016, 74% in 2017, 62% in 2018, and 68% in 2019 of all admitted patients to the ED unit. The Psychiatric Center Ballerup (PCB) is the only ED unit for treatment of ED in the capitol region of Denmark, having a catchment area of 2 million inhabitants and an expected prevalence of ED of about at least 25,000.

The diagnosis of ARFID was made according to the ICD-11 criteria [1], and the diagnosis of AN was made in accordance with the ICD-10 criteria at the time of investigation (International Statistical Classification of Diseases and Related Health Problems, 1992). All patients were recruited through their referral to the PCB.

All patients were undergoing a weight restoration program, either as inpatients or as taking part in an intensive day care treatment (4 weekdays of 5). Meals were provided 3 (daycare) to 5 (inpatient) times per day and food intake was monitored by trained nurses. A dietician planned the meals individually to enable an approximate 1 kg weight increase per week up to a ideal body weight (IBW) defined as BMI 20 for women and BMI 21 for men. Weight gain was supported by restrictions in physical activity, monitored meals and post-meal rest. Weekly measures of weight were done, and all patients had undergone medical and psychiatric examinations and any medical complications had been addressed. There was no formal psychotherapy provided during this time period while individual meetings with psychologists and nurses was offered to the patients. Meal groups with cooking skills was also provided by dieticians. All patients received vitamins, however, no patient in this study had undergone enteral feeding during this course of treatment. Body awareness therapy was given by trained physiotherapists. An average stay for the ED unit was 10 weeks independent of cause for discharge. The individuals included in this analysis had all decided to remain in the PROLED study, while some may have dropped out of clinical treatment during the 12 weeks of follow-up.

### 2.2. Clinical Measures

A complete diagnostic work-up including a comprehensive diagnostic interview by a psychologist, medical and psychiatric examinations carried out by either a specialist psychiatrist, or a General Practitioner with special training in Eating Disorders. Assessment was accomplished using the Eating Disorder Examination (diagnostic questions; EDE [17]) and routine clinical and laboratory assessments to ensure high quality diagnosing of ED and comorbid disorders. All primary and comorbid diagnoses except a diagnosis of ARFID, were validated by a second, independent Physician, using the ICD-10 checklist [18]. ARFID was diagnosed by the primary physician, then evaluated by two independent physicians checking the ICD-11 diagnostic criteria. Cohens kappa was calculated as 0.86 for a diagnosis of ARFID vs. AN-R (http://vassarstats.net/kappa.html, accessed on 7 March 2021). The PROLED study includes validated questionnaires that assess general and specific aspects of ED, as well as comorbid disorders such as depression, anxiety, personality, previous trauma, autism, cognition, social aspects, global function, quality of life and psychotic features. For this study, the Major Depressive Inventory (MDI), a 10 item self-assessment, high external validity instrument to assess depression symptoms and depth, with item scores ranging from 0–5 on a likert scale, providing a sum score where higher scores indicates more severe depression, [19] was used for depression assessment. The Eating Disorder Examination Questionnaire (EDE-Q), developed from the Eating Disorder Examination [20], which consists of 32 items assessing the frequency and severity was used to assess ED symptoms. The EDE-Q also yields four subscales from twenty-three items: Dietary Restraint, Eating Concern, Weight Concern, and Shape Concern. Both the two-week test-retest reliability (coefficients ranging from 0.81 to 0.92) and internal consistency (alphas ranging from 0.78 to 0.93 [21]) for EDE-Q have been found high [22]. The Autism Quotient (AQ), a 50 item self-report questionnaire for adults, divided into five subscales consisting of 10 items each, was used to asses domains of cognitive strengths and difficulties related to ASD: communication, social skills, imagination, attention to detail and attention switching [23]. The Hopkins Symptom Check List 92 (SCL-92), a 92 item self-assessment instrument where each item is ranked in 5 levels from “totally agree” to “not agree at all”, was used to asses different psychiatric symptoms such as: somatization, obsession/compulsion, interpersonal sensitivity, depression, anxiety, hostility, phobic anxiety, paranoid ideation and psychoticism, as well as a total score for overall distress of psychiatric disease [24]. The Morgan Russell Outcome Scale (MROS) was used to assess additional symptoms associated with ED, and which includes five domains (eating difficulties, menstrual state, mental state, psychosexual state, socioeconomic state), in which mean scores generates a composite rating on a 12-point scale where high scores indicate a good prognosis [25]. In addition, the Standardized Assessment of Personality Abbreviated Scale (SAPAS), which consists of eight questions that corresponds to a descriptive statement about the person, scored as 0 or 1 and added together produce a total score between 0 and 8 with higher score reflecting likelihood of a diagnosis of personality disorder [26]. Finally, the readiness ruler, an 18-item self-questionnaire on readiness to change, which includes 9 domains: restriction, weight, shape overvaluation, binge eating, vomiting, laxative use, fasting, diuretic use, weight-gain phobia and exercise, and has two sections; Likert scale items 1–9 on readiness to change, and, 0–100% rating of motivation to change. The same nine behaviors are rated for being made for others versus for oneself. The readiness ruler total scores are provided for readiness and autonomy to change [15]. All these instruments were used to characterize the enrolled patients at baseline and explore potential differences on a group level. We only used validated Danish versions of the instruments except readiness ruler, which as yet only exists as a translated version.

Weight was assessed weekly and difference from baseline and BMI was calculated at 6 weeks (w), 8 w, 10 w and 12 w. (https://www.nhlbi.nih.gov/health/educational/lose_wt/BMI/bmicalc.htm, accessed on 7 March 2021).

### 2.3. Statistical Analyses

All analyses were conducted using the SPSS Version 25., IBM, New York, NY, USA. The distribution of the included data was assessed using the Kolmogorov-Smirnoff test and any outliers and/or missing data were excluded by case. Since the data were not normally distributed, baseline characteristics were compared between ARFID and AN-R using the Mann–Whitney U-test. Correlations between BMI and questionnaire scores at baseline was done using Spearman Rank correlation test for all included individuals to explore potential associations between these factors at baseline to BMI. Follow-up assessments for changes in BMI between the ARFID and AN-R were analyzed using the Mann–Whitney U-test of differences in BMI from baseline to week 6, w8, w10, and w12 of weight restorative therapy. Chi-2 was used to analyze differences between ARFID and AN-R in the frequencies of psychiatric comorbidities. All tests were 2-tailed, and correction for multiple testing was done using Bonferroni correction setting the *p*-value at 0.0125 for the main analyses (change in BMI).

## 3. Results

### 3.1. Comparison of ARFID and AN-R groups at baseline

The descriptive data of the sample selected for this study is summarized in Table 1. Baseline anxiety as measured with SCLD16 and EDE-Q, differed significantly both in the total score and in all sub-scores between ARFID and AN-R. Apart from that, there were no additional differences in baseline characteristics between the two groups. In addition, no differences in the frequencies of psychiatric comorbidities between ARFID and AN-R were found (Depression ARFID 25%, AN-R 16%; Anxiety syndromes ARFID 12%, AN-R 11%; Obsessive Compulsive disorder ARFID 25%, AN-R = 9%).

### 3.2. Comparing Weight Change between ARFID and AN-R Groups

Comparing ARFID and AN-R regarding weight change after 6 weeks (w), 8 w, 10 w and 12 w from baseline, no differences were found (Table 2).

Using Spearman’s rho, Baseline BMI was correlated to SCLA-14 (SCL Anxiety scale 14 items) score (R = 0.42; *p* = 0.002; *n* = 49), SCLD-16 (R = 0.36; *p* = 0.01; *n* = 49; SCL Depression scale 16 items). In addition, age was correlated to SAPAS score (R = 0.34; *p* = 0.01; *n* = 55). Other correlations were identified as non-relevant since reflecting the interrelation between the same variables in different questionnaires. For ARFID case presentation, see Table 3.

## 4. Discussion

The main finding of this study was that there was no difference between ARFID and AN-R regarding weight gain during the first six weeks up to 12 weeks of weight restoration treatment. This finding resembles what other scientists have found in previous studies in children and adolescents [6,7,8]. Our hypothesis of a potential slower weight gain in ARFID was not corroborated by the findings. Instead, patients with ARFID gained weight from the restorative treatment at the same rate as patients with AN-R. This indicates that the standard weight restorative treatment program designed for patients with AN-R is effective also in underweight individuals with ARFID, and there would be no need for an personalized program for this diagnostic group, at least when provided under clinically strictly controlled circumstances such as in an inpatient setting. The uniqueness of this study with frequent measures of weight gain, the prospective design and including only adults, makes it the first study of its kind. Other studies conducted in adults have frequently been retrospective reviews of medical records [4,6,11] and suffered from shorter follow-up time [4,6], or have included both children and young adults [3] making comparisons with our study difficult due to differences in study designs. Studies in children and adolescents have described the ARFID group to be younger (at onset), have a higher heredity and comorbidity for mental disorders [10], findings which are not corroborated in the current study on adults. However, previous studies have stressed an association to ASD [12,13,14], which we only found in one of the seven enrolled individuals. We did however find fear of nausea and vomiting, and attention to details or characteristics in food frequently in the ARFID cases. These signs, together with food avoidance and being picky eaters, may indicate some undiagnosed overlap with ASD in these individuals due to its resemblance with the cognitive rigidity frequently found in ASD [27,28].

Baseline anxiety was higher in AN-R than in ARFID. Several other studies have reported on a higher frequency of psychiatric comorbidity in ARFID [29], especially anxiety [30,31,32]. In the current study, although the anxiety score was higher in AN-R, no difference in comorbidities were found. In view of the purpose of this current study, it seems that this increased anxiety in ARFID did not influence weight gain differently from that in AN-R, despite any baseline differences, weight gain at the various time points were similar between ARFID and AN-R, casting doubt on the potential influence on baseline scores as moderators of weight increase in the studied disorders.

At baseline, depression, anxiety and personality scores were correlated with BMI, when including all individuals. The latter was done to increase power of the analysis. These associations may point to that BMI influenced affective and personality scores in restrictive ED patients. However, individual assessment would be informative and is lacking. This needs to be explored in future studies including follow-up assessments.

Baseline scores of EDE-Q, both global and sub-scores, differed between ARFID and AN-R. This finding is explained by the, per definition, more severe ED pathology in AN-R as reflected in the differences in the diagnostic criteria. Patients with ARFID most notably do not have a disturbance in their body image. In spite of this difference in psychopathology, this does not influence or lead to any differences in the rate of weight gain when under strictly controlled inpatient care and subjected to weight restorative treatment. Since the current study only followed weight gain in the short term, any differences in the long term, e.g., in the number of relapses and the long-term stability in weight after restorative treatment, are currently unknown and will need to be assessed in follow-up studies.

The fact that there is no difference in weight gain from weight restorative treatment between ARFID and AN-R, supports inpatient and daycare efforts to try to reduce the medical and psychiatric risks associated with these diagnoses. The AQ scores at baseline, including information from the medical records, indicated that autistic traits were more frequent in the ARFID cases, which also is consistent with case reports [33] and clinical studies [34]. Although not part of this study, among clinicians there is frequently an impression that patients with ASD and eating disorders have more difficulties in gaining weight and drop out earlier from treatment (personal communication clinicians at PCB). This clinical impression could not be supported by the current findings.

## 5. Limitations

This study suffered from the fact that only a few cases of ARFID available for comparison. Including a larger number of individuals with ARFID would increase the statistical power of study and the validity of the findings. However, the sample represents all cases of ARFID that were enrolled in the PROLED study during a 5-year period, and overall, the enrollment rate into PROLED of all admitted patients to the ED unit was high. In addition, since the unit is the only specialized ED unit for adults in the region, having approximately 2 million inhabitants, our sample is most likely representative of the adult ARFID cases in need of inpatient weight restorative treatment in this catchment region, which is a strength of this study. Another limitation is that the diagnoses of ARFID was made based upon clinical interviews and retrospective validation, but not on standardized ARFID specific clinical interviews. However, a strength is that all these included cases clearly fulfilled both the ICD-11 and the DSM-5 criteria for a diagnosis of ARFID. Furthermore, a thorough review of medical records showed support for long term findings consistent with a diagnosis of ARFID and the fact that the scores of EDE-Q, both the global and the sub-scores, differed from that of AN at baseline supports the validity of the diagnosis of ARFID and thereby the representativeness of these cases for this diagnosis.

Due to the small number of ARFID cases included, adjusting for baseline differences was not possible. Further analysis in a larger set of patients with an ARFID diagnosis will enable adjusting for potential confounders such as for example, the duration of treatments, the number of dropouts, potential differences in motivation for treatment, and differences in severity of anxiety and eating disorder psychopathology, into consideration.

## 6. Conclusions

The current study finds no difference in weight gain between adult individuals with ARFID and AN-R thus refuting our hypothesis. Instead, with the cases included in this study, weight gain from inpatient or day care treatment in ARFID seems as effective as that in AN-R. This study also finds baseline differences in several clinical variables, which warrants further exploration in the larger set of patients.

## Figures and Tables

**Table 1 nutrients-13-00871-t001:** Baseline clinical characteristics.

Characteristics	ARFID *N* = 7	AN-R *N* = 80
Age in years	24 (20–45)	22 (18–53)
Duration in years	10.5 (1–22)	6.5 (0–34)
Gender	5 women	77 women
Baseline BMI	14.3 (12.7–17.4)	15.1 (11.4–17.9)
SCL-Mood score	0.9 (0.7–1.9)	1.9 (0.4–3.2)
SCL-anxiety score	1.0 (1.0–1.3)	2.2 (0.6–3.5) **
AQ	15 (0–30)	15 (0–40)
MROS	6.8 (4.7–9.3)	5.6 (1.8–11.6)
SAPAS	3 (2–4)	4 (0–7)
Readiness Ruler	3.8 (1.1–5.1)	3.8 (2.0-5.9)
EDEQ-Restraint	0.0 (0.0–3.0)	3.2 (0–6) ***
EDEQ-Eating Concern	0.9 (0.0–4.0)	3.4 (0–6) **
EDEQ-Shape Concern	1.7(0.0–4.0)	4.9(0–6) ***
EDEQ-Weight concern	1.6(1.0–4.0)	4.4(0–6) **
EDEQ-total	1.3(0.0–4.0)	4.1(0–6) ***

Statistically significant group differences are indicated ** <0.01, *** <0.001. Median with range within parentheses if not otherwise specified. Abbreviations: AN-R = Anorexia Nervosa Restrictive type; ARFID = Avoidant Restrictive Food Intake Disorder; AQ = Autism Quotient; BMI = Body Mass Index; EDE-Q = Eating Disorder Examination Questionnaire; MROS = Morgan-Russell Outcome scale; N = number of individuals; SAPAS = Standardized Assessment of Personality Abbreviated Scale; SCL = The Hopkins Symptom Check List.

**Table 2 nutrients-13-00871-t002:** Mean change from Baseline in BMI.

BMI	ARFID *N* = 7	AN-R *N* = 74	Mann–Whitney U test *p*-Value
6 weeks	1.4 (0.8–1.8)	1.2 (−1.1–3.8)	0.163
8 weeks	1.9 (0.7–2.5) ^	1.6 (−1.1–4.8) ~	0.488
10 weeks	2.2 (1.1–3.6) ^^	2.0 (−0.9–4.1) ~~	0.464
12 weeks	3.2 (0.7–4.4) ^^^	2.7 (−1.3–4.8) ~~~	0.431

No changes significantly different. ^ *N* = 6, ^^ *N* = 5, ^^^ *N* = 4, ~ *N* = 68, ~~ *N* = 61, ~~~ *N* = 54. Median with range within parentheses if not otherwise specified. Abbreviations: AN-R = Anorexia Nervosa Restrictive type; ARFID = Avoidant Restrictive Food Intake Disorder; BMI = Body Mass Index; N = number of individuals; w = weeks.

**Table 3 nutrients-13-00871-t003:** Case presentation of ARFID participants. All patients fulfilled the ICD-11 and DSM-5 criteria.

Age/Gender	20/F	21/F	22/F	26/F	45/F	22/M	26/M
BMI start	13.8	14.1	16.6	17	12.8	14.5	17.4
BMI end	16	In treatment	21.1	19.7	16.4	13.8	18.9
Weight change	+5.8 kg		+11.1 kg	+8 kg	+10.7 kg	−2.1 kg	+4.7 kg
Duration of treatment and setting	7 w. inpatient	16 w. inpatient	15 w. inpatient	9 w. inpatient	16 w. inpatient	58 w. daycare	8 w. inpatient
Psychiatric comorbidities	OCD Anxiety	Anxiety	Schizotypal PD. Anxiety/depressive symptoms Previous self-harm	OCD Anxiety/depressive symptoms	Panic disorder	Atypical autism Gaming disorder	ADHD Unipolar depression Somatoform syndrome Anxiety
Medical comorbidities	GERD Hypokalemia	Congenital neutropenia Lactose intolerance	Osteopenia Migraine w. aura	Osteopenia IBS Hyposalivation	Vitamin B deficiency Poor dental status	Osteopenia Dyslexia Poor dental status	None
Family history	None	Hyperlipidemia TCI	Stress Depressive symptoms Suspected Eating disorder	Alcohol abuse	Breast cancer	Autistic traits	ADHD Depression Fibromyalgia Asperger’s Tourette’s
Education	Upper secondary school	Upper secondary school	Upper secondary school	10th grade exam	Higher education completed	Upper secondary school	10th grade exam
Duration of ED (years)	3 years (always low weight)	9 years (restrictive eating)	10 years (restrictive eating)	12 years (restrictive eating)	22 years (food avoidance)	17 years (food avoidance)	1 year (always low weight)
Presentation on admission	Significant weight loss and increasingly restrictive eating pattern	Weight loss during outpatient ED care	Weight loss during outpatient ED care	Weight loss of 4.9 kg (4 w) Restrictive eating	Need for intensified treatment	Need for intensified treatment	Weight loss during outpatient ED care
Characteristics	Food avoidance. Fear of vomiting and reflux from eating. Picky eater	Food avoidance. Fear of nausea, reflux and abdominal pain. Perfectionistic traits.	Food avoidance. Fear of negative emotions due to cognitive enhancement from eating. Perfectionistic traits.	Food avoidance (food characteristics) Fear of abdominal pain and negative feelings from eating. Picky eater.	Food avoidance. Fear of vomiting. Picky eater.	Food avoidance (food characteristics) Fear of abdominal pain and vomiting. Lack of interest in eating. Picky eater.	Food avoidance. Fear of vomiting, abdominal pain and choking. Picky eater.
Sensory sensitivity				yes		yes	
Fear of aversive consequences	yes	yes	yes	yes	yes	yes	yes
Lack of interest						yes	
Compensatory behavior	None	None	None	None	None	None	None
Substance abuse	None	None	None	None	None	None	None

## Data Availability

The datasets used in this study are available from the corresponding author.

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
