# Peer review of "Weight Gain in Adults with Avoidant/Restrictive Food Intake Disorder Compared to Restrictive Anorexia Nervosa—Pilot Findings from a Longitudinal Study"

_nutrients, 2021, doi:10.3390/nu13030871_

Round 1

Reviewer 1 Report

The study aims to prospective explore possible differences in weight gain and psychopathological comorbidities between adults diagnosed with ARFID and AN-R. Although the focus is very interesting, the sample size is clearly too small (only 7 adults with ARFID Compared to 80 with AN-R) to meet routine power requirements.

Please find below some comments.

ABSTRACT

The abstract must clearly state the aims of the study. The authors refer only to wanting to explore differences in weight gain. Only in the results they mention the presence of differences in psychopathological symptoms associated to eating disorders.

INTRODUCTION

The introduction is well written but fairly biased.

The authors should better clarify what the rationale for the study is.

Why is this study important? What does it add over previous literature?

In addition, it lacks a clear theoretical framework, which should be made explicit from the introduction and also taken up in the discussions of the results.

In addition, it is important to more clearly describe the objectives of the study and related hypotheses, including theoretical ones, based on previous literature.

METHODS

Many important information is not described.

What are the criteria for sample inclusion/exclusion?

Did participants sign an informed consent form?

Has the study been accepted by the ethics committee? What is the protocol number?

It is necessary to provide  the degree of agreement (k cohen) between the two independent physicians that made diagnosis.

More information needs to be provided with respect to all instruments administered (EDE, MDI, AQ, etc.). What do they evaluate? How many items do they consist of? Psychometric characteristics? Have you used a validated version for the Danish population?

One of the major limitations of the study concerns the sample. In addition to the very small number of the sample with ARFID diagnosis (7 subjects vs 80 subjects with AN-R), the two groups were not balanced with respect to gender either. The sample with AN-R in fact appears to be composed almost exclusively of men.

How have you considered these important limitations at the statistical level?

It would perhaps be more appropriate to refer to the study in the terms of a pilot study

DISCUSSION

Overall, the authors discuss each results based on previous studies. They evidence significant associations emerged between variables, but without providing theoretical hypotheses underlying these associations.  The theoretical framework from which the authors start for their study needs to be clearer and more explicit. Consistently, I suggest to enrich the discussion with theoretical references that support the hypothetical links between the variables emerge from the study.

Author Response

Response to reviewer

nutrients-1077835, Weight gain in adults with avoidant/restrictive food intake disorder compared to restrictive anorexia nervosa – findings from a longitudinal study

reviewer 1

Comments and Suggestions for Authors

The study aims to prospective explore possible differences in weight gain and psychopathological comorbidities between adults diagnosed with ARFID and AN-R. Although the focus is very interesting, the sample size is clearly too small (only 7 adults with ARFID Compared to 80 with AN-R) to meet routine power requirements.

  • We thank this reviewer for constructive feedback. Please see the rationale for the sample size in one of our responses below.

Please find below some comments.

ABSTRACT

The abstract must clearly state the aims of the study. The authors refer only to wanting to explore differences in weight gain. Only in the results they mention the presence of differences in psychopathological symptoms associated to eating disorders.

  • The aim has been rewritten to more clearly state the aim and now reads as follows:
    “Weight restorative treatment in adults is provided as in Anorexia Nervosa (AN), while the effect in unknown. The aim was to compare weight gain, including exploring influencing factors, in ARFID to restrictive subtype of AN (AN-R).

INTRODUCTION

The introduction is well written but fairly biased.

  • We are unsure if this is a statement or a question. The introduction is written to harmonize with the aim of the study which naturally tends to be a selection of all potential topics.

The authors should better clarify what the rationale for the study is.

  • The aim in the end of the introduction has been reworded to more clearly state the aim and now reads as follows “In view of the few studies executed investigating weight gain in ARFID, and the heterogeneity of these, there is a need for evaluating the effect of weight restorative treatment typically provided to patients with AN, on weight gain in ARFID. Consequently, the aim of the current study was to compare the effect of weight restorative treatment in ARFID compared to AN-R, including exploring baseline factors that may influence weight gain during inpatient treatment.”

Why is this study important? What does it add over previous literature?

  • The importance of this publication is presented in the end of the introduction and reads as follows:
    “In view of the few studies executed investigating weight gain in ARFID, and the heterogeneity of these, there is a need for evaluating the effect of weight restorative treatment typically provided to patients with AN, on weight gain in ARFID.. “

    and we have added the following text to provide further rationale for the study:
    “ Any disparate findings may influence treatment programs for ARFID. “

In addition, it lacks a clear theoretical framework, which should be made explicit from the introduction and also taken up in the discussions of the results.

  • We have added a theoretical framework supporting our hypothesis, in the introduction, which now reads as follows:

ARFID is a relatively newly described disorder and due to the low body weight and the restrictive food intake, may be mistaken for Anorexia Nervosa (AN) which is, in contrast to ARFID, characterized by phobia of weight increase, low body weight, body image distortion and compensatory behaviours such as frequent exercise. Few studies have examined the effects of weight restorative treatment in ARFID. A systematic literature search using PUBMED on weight gain or weight increase in ARFID identified 19 studies whereof only 8 compared weight gain in ARFID to AN, and none included adults only but children to young adults (mean age 23,5 years; in total 188 ARFID cases). Some studies found a slower weight gain in ARFID compared to AN [3,4], and one study a lesser weight gain in ARFID [5], while a few other studies did not find any significant differences in weight gain [6-8]. Background or influencing factors were not systematically investigated and the follow-up time varied between 1 week [3,6] to 3-4 weeks [3,4] and 10 weeks [5,9,10] or more [11]. The rationale for a slower and lesser weight gain in ARFID may relate to the differences in psychopathology, individuals with ARFID frequently sharing traits with autism [12-14] where rigid principles may interfere with treatment approaches that are developed from experienced with treatment of Anorexia nervosa and based upon supporting intrinsic motivation, factors that may be reflected at baseline assessments for example in readiness to change. This may also explain why individuals with ARFID more often have a chronically low weight compared to AN [15]

In addition, it is important to more clearly describe the objectives of the study and related hypotheses, including theoretical ones, based on previous literature.

  • See responses to previous questions.

METHODS

Many important information is not described.

What are the criteria for sample inclusion/exclusion?

  • Inclusion criteria have been added to the manuscript and now reads:

“The following individuals are eligible for enrollment subject to signed written informed consent:

-adult individuals (age at or above 18 years – 65 years)

-admitted to the ED unit in Psychiatric Center Ballerup, Denmark

- a diagnosis of an eating disorder”

Did participants sign an informed consent form?

  • This information has been added, under Material and methods 2.1.

Has the study been accepted by the ethics committee? What is the protocol number?

  • Yes, protocol id has been added, under Material and methods 2.1. and now reads as follows:
    “It has been approved by the local ethics board (id: H-15012537; addendum 77106) and the data processing board”

It is necessary to provide  the degree of agreement (k cohen) between the two independent physicians that made diagnosis.

  • This has been added under 2.2, and was calculated as 0.86 for a diagnosis of ARFID and AN-R. http://vassarstats.net/kappa.html

More information needs to be provided with respect to all instruments administered (EDE, MDI, AQ, etc.). What do they evaluate? How many items do they consist of? Psychometric characteristics? Have you used a validated version for the Danish population?

  • We have added information on all included instruments under section 2.2 and also information that we only used validated Danish versions of the scales.

One of the major limitations of the study concerns the sample. In addition to the very small number of the sample with ARFID diagnosis (7 subjects vs 80 subjects with AN-R), the two groups were not balanced with respect to gender either. The sample with AN-R in fact appears to be composed almost exclusively of men.

  • The sample represents all cases that were enrolled in the PROLED study during a 5-year period. We have added this information in the abstract and in section 3.1. The strength of this study thereby is that it represents all cases of ARFID enrolled during 5 years. For all included individuals with an ED diagnosis, including AN-R, the enrollment rate in 2016 was 90% of all admitted patients, and in 2018 it was 80%. Thereby, the selection bias is negligible. We agree that it would have been advantageous to have a larger sample, however, as the Eating disorder unit is the only ED specialized unit in the region of 2 million inhabitants, for adults, we believe that our sample is representative of the rate of referrals of ARFID cases in this catchment region. In addition, the inclusion of other cases would have required a specific recruitment e.g. via announcements, which thereby would have led to a clear selection bias.

How have you considered these important limitations at the statistical level?

  • Yes and we have added a section on limitations in the discussion.

It would perhaps be more appropriate to refer to the study in the terms of a pilot study

  • Since this study was a longitudinal study that span over 5 years and all admitted patients with an ED diagnosis was invited to participate, and since the participation rate in 2016 was 90%, and 80% in 2018, we prefer to describe this study as a “longitudinal study” i.e. the PROLED study.

DISCUSSION

Overall, the authors discuss each results based on previous studies. They evidence significant associations emerged between variables, but without providing theoretical hypotheses underlying these associations.  The theoretical framework from which the authors start for their study needs to be clearer and more explicit. Consistently, I suggest to enrich the discussion with theoretical references that support the hypothetical links between the variables emerge from the study.

  • We agree with the reviewer and have enriched the discussion and added a theoretical framework and references.

Reviewer 2 Report

This is an important subject given that adult services are only recently beginning to see numbers of ARFID patients  A significant problem, acknowledged by the authors, is the very small number of ARFID patients which makes the study fall between a case series and a cohort study.  The only remedy would be to collect a larger sample, perhaps in a multi-centre study.  Secondly, the authors state in the abstract: "The differences in EDE-Q stresses that AN-R patients have a more severe eating disorder psychopathology than ARFID patients." This is true, but necessarily arises from the different definitions of AN and ARFID, so is expected. I would think that the authors might allude to this in their discussion. 

Author Response

reviewer 2

Comments and Suggestions for Authors

This is an important subject given that adult services are only recently beginning to see numbers of ARFID patients 

  • We thank this reviewer for constructive feedback and agree that services for adults with ARFID, also in Denmark are currently not specified.

A significant problem, acknowledged by the authors, is the very small number of ARFID patients which makes the study fall between a case series and a cohort study.  The only remedy would be to collect a larger sample, perhaps in a multi-centre study. 

  • The sample represents all cases that were enrolled in the PROLED study during a 5-year period. We have added this information in the abstract and in section 3.1. The strength of this study thereby is that it represents all cases of ARFID enrolled during 5 years. For all included individuals with an ED diagnosis, including AN-R, the enrollment rate in 2016 was 90% of all admitted patients, and in 2018 it was 80%. Thereby, the selection bias is negligible. We agree that it would have been advantageous to have a larger sample, however, as the Eating disorder unit is the only ED specialized unit in the region of 2 million inhabitants, for adults, we believe that our sample is representative of the rate of referrals of ARFID cases in this catchment region. In addition, the inclusion of other cases would have required a specific recruitment e.g. via announcements, which thereby would have led to a clear selection bias.

Secondly, the authors state in the abstract: "The differences in EDE-Q stresses that AN-R patients have a more severe eating disorder psychopathology than ARFID patients." This is true, but necessarily arises from the different definitions of AN and ARFID, so is expected. I would think that the authors might allude to this in their discussion. 

  • We agree with this reviewer and have removed the statement from the abstract, and added a section in the discussion which reads as follows:

  • “Baseline scores of EDE-Q differed between ARFID and AN-R. This finding is explained by the, per definition, more severe ED pathology in AN-R as reflected in the differences in the diagnostic criteria. Patients with ARFID most notably do not have a disturbance in their body image. In spite of this difference in psychopathology, this does not influence or lead to any differences in the rate of weight gain when under strictly controlled inpatient care, and subjected to weight restorative treatment. Since the current study only followed weight gain in the short term, any differences in the long term e.g. in the number of relapses and the long-term stability in weight after restorative treatment, are currently unknown and will need to be assessed in follow-up studies. “

Round 2

Reviewer 1 Report

The authors have moficated the manuscript in accordance with the comments I placed. In this version, I believe the article can be accepted for publication

Author Response

We are grateful to this reviewer for her/his valuable comments and feedback, which all has improved the manuscript.

Reviewer 2 Report

Thank you for making the changes which appear appropriate. Just one small change "Morgan-Russel" should be "Morgan-Russell".  Gerald Russell was quite perfectionistic and would not approve of the mis-spelling. 

Author Response

We are grateful to this reviewer for her/his valuable comments and feedback, which all has improved the manuscript. we have updated the manuscript to ensure spelling of Morgan-Russell is correct throughout the text.